# Bio-Based Flame-Retardant Systems for Polymers Obtained via Michael 1,4-Addition

**DOI:** 10.3390/molecules30122556

**Published:** 2025-06-11

**Authors:** Kamila Salasinska, Mateusz Barczewski, Mikelis Kirpluks, Ralfs Pomilovskis, Paweł Sulima, Sławomir Michałowski, Patryk Mietliński, Jerzy Andrzej Przyborowski, Anna Boczkowska

**Affiliations:** 1Faculty of Materials Science and Engineering, Warsaw University of Technology, Wołoska 141, 02-507 Warsaw, Poland; anna.boczkowska@pw.edu.pl; 2Institute of Materials Technology, Poznan University of Technology, Piotrowo 3, 61-138 Poznan, Poland; mateusz.barczewski@put.poznan.pl; 3Polymer Laboratory, Latvian State Institute of Wood Chemistry, 27 Dzerbenes St., LV-1006 Riga, Latvia; mikelis.kirpluks@kki.lv (M.K.); ralfs.pomilovskis@kki.lv (R.P.); 4Department of Genetics, Plant Breeding and Bioresources Engineering, University of Warmia and Mazury, Plac Łódzki 3, 10-724 Olsztyn, Poland; pawel.sulima@uwm.edu.pl (P.S.); jerzy.przyborowski@uwm.edu.pl (J.A.P.); 5Department of Chemistry and Technology of Polymers, Cracow University of Technology, Warszawska 24, 31-155 Cracow, Poland; slawomir.michalowski@pk.edu.pl; 6Institute of Mechanical Technology, Poznan University of Technology, Piotrowo 3, 61-138 Poznan, Poland; patryk.mietlinski@put.poznan.pl

**Keywords:** plant fillers, chemical composition, intumescent flame retardant, thermal properties, flammability

## Abstract

Phosphorus flame retardants react with cellulose hydroxyl groups via esterification, enhancing the effectiveness of char formation, which is beneficial in terms of the search for bio-sourced flame retardants. The current work assessed the flammability of a new polymer synthesized by Michael 1,4-addition (rP) and modified with developed intumescent flame retardant systems (FRs), in which lignocellulose components, such as sunflower husk (SH) and peanut shells (PS), replaced a part of the synthetic ones. The thermal and thermomechanical properties of the rP, with 20 wt.% each from six FRs, were determined by thermogravimetric analysis (TG), differential scanning calorimetry (DSC) and dynamic mechanical thermal analysis (DMTA). Moreover, the flammability and evolved gas were studied with pyrolysis combustion flow calorimetry (PCFC) and thermogravimetric analysis connected with Fourier transform infrared spectroscopy tests (TGA/FT-IR). The effects were compared to those achieved for unmodified rP and a polymer with a commercially available intumescent flame retardant (IFR). The notable improvement, especially in terms of the heat release rate and heat release capacity, indicates that the system with melamine phosphate (MP) and peanut shells (PS) can be used to decrease the flammability of new polymers. An extensive analysis of the composition and geometry of the ground shells and husk particles preceded the research.

## 1. Introduction

Due to legal regulations (REACH regulation, Stockholm Convention on Persistent Organic Pollutants, EU RoHS Directive on the restriction of the use of certain hazardous substances in electrical and electronic equipment [1,2,3]), rapidly rising prices and shortages of products caused by the global economic and political situation, as well as the devastation of the natural environment, the attention of researchers has become directed to the search for new sustainable solutions in polymer technology, including the area of fire retardancy. Recent efforts to develop new flame retardants (FRs) have shifted strongly toward phosphorus-based and other halogen-free systems. Phosphorus FRs can remain in the solid phase and promote charring or volatilization, acting as H- or OH-radical scavengers. Volatile phosphorus compounds are among the most effective combustion inhibitors, being five times more effective than bromine and ten times more effective than chlorine. They are more effective in oxygen- or nitrogen-containing materials, which can be either heterochain polymers or polymers with proper functional groups due to the condensed-phase mode of action [4]. Unfortunately, the halogen-free FRs available on the market are often ammonia-based compounds or those with a high content of aromatic rings, which are obtained through multi-stage processes that require significant amounts of hazardous substances.

The fire retardancy potential of plant materials is rooted in their chemical structure and possible interactions with other decomposition products of polymers modified with synthetic FRs. Lignocellulosic materials contain many polar groups relevant to the effectiveness of phosphorus-based FRs, including high concentrations of hydroxyls and carboxyls [5]. Phosphorus FRs react with cellulose hydroxyl groups via esterification or transesterification, and as a result, a char is formed at the expense of combustible volatile products. Additionally, compounds such as urea, dicyandiamide, and melamine accelerate the phosphorylation of cellulose by forming a P–N intermediate and thus synergize the flame retardant action of phosphorus. Usually, instead of cellulose, a chemical compound such as pentaerythritol (polyol) is added, which, together with the acidic promoter and melamine, forms a multicellular and swollen char, providing the expected temperature gradient. The char impedes the heat flux and retards the diffusion of volatile pyrolysis products to the flame [4,6]; therefore, it is considered one of the most effective passive protection methods and has been used even in wooden and steel structures.

There are several examples of the employment of bio-based FR systems reported in the literature [7,8,9,10]. Battig et al. [5] uutilized leather waste (LW) modified with melamine-encapsulated ammonium polyphosphate (APP) as a multifunctional bio-filler, thereby enhancing the fire performance of poly(ethylene–vinyl acetate) (EVA). The improved char stabilization of EVA reduced the heat release rates (HRR) by up to 53% compared to polymers and up to 40% compared to EVA modified with commercial IFR. APP interacted with the polar groups from LW, including hydroxyl groups from hydroxyproline and carboxylic acids. The abundance of reactive sites increased the presence of phosphorus in all EVA/LW moieties, which acted as cross-link points between the decomposing collagen from LW and the EVA. As a result, residue retention increased, and phosphorylation via transesterification occurred; meanwhile, the polymer underwent dehydration and cyclization reactions. Moreover, the formation of P-N compounds (phosphorus oxynitrides, phosphazenes) increases charring and stability. In turn, ammonia acts as a blowing and flame dilution agent.

Meanwhile, as reported by Salasinska et al. [11], using nutshells and sunflower husks for epoxy resin (EP) modification resulted in a reduction in the critical parameters used to assess the flammability of polymers. It was noticed that the size and quality of the char were affected by differences in the complex structure of the bio-based components (amount of lignocellulose and hemicellulose). Char formation also impacted smoke emissions. The most significant reduction (2.5 times compared to unmodified EP) was recorded for hazelnut shells, which, similar to sunflower husk, achieved the most significant reduction in the total amount and variety of toxic burning products emitted. An alternative solution implemented in our proceeding study was the design of a halogen-free flame-retardant system containing a hazelnut shell. This solution resulted in even greater effectiveness. The use of 5 wt.% of nutshells in combination with 15 wt.% L-histidine dihydrogen phosphate-phosphoric acid (LHP) contributed to a 6-fold decrease in heat release due to the formation of the desired char structure. The analysis confirmed the synergistic effect between components [12].

The selection of polymers synthesized via the 1,4-Michael addition reaction is grounded in the need for sustainable alternatives to conventional polymers. In line with the European Green Deal’s targets for climate neutrality by 2050, recent research has emphasized the valorization of renewable feedstocks, such as rapeseed oil [13], tall oil [14], and lignocellulosic industry by-products [15], to develop bio-based monomers suitable for polymer synthesis. The 1,4-Michael addition offers a versatile, base-catalyzed route for the reaction of β-ketoester-functionalized Michael donors with multifunctional acrylate acceptors under mild conditions. This approach eliminates the use of toxic components, such as isocyanates, enables the design of thermoset polymers with tailored cross-linking densities, and yields materials with thermal and mechanical properties comparable to conventional systems.

As part of this study, six bio-sourced retardant systems for the new polymer were developed using the Michael addition, resulting in reduced flammability. The polymer was made from an aromatic polyester polyol obtained from industrial PET waste, and its characteristics, including the structure of the polymer and manufacturing methods, have been described in previous works [14,16]. The acetoacetate-modified Neopolyol 380 was chosen as a feedstock for this study when developing the materials with reduced flammability as it offered competitive mechanical properties, such as Young’s modulus above 2.4 GPa [14], and sustainability as it is sourced from recycled PET. The influence of the developed FRs on the chemical structure of the polymer was determined using Fourier transform infrared spectroscopy (FT-IR). Next, each material containing 20 wt.% of the FR system was subjected to a series of thermal (DSC, VST) and mechanical (hardness, impact strength, DMA) investigations, and the evaluation of the properties were related to the microstructure analysis. Research techniques, including thermogravimetric analysis (TGA), pyrolysis combustion flow calorimetry (PCFC), and thermogravimetric analysis coupled with Fourier transform infrared spectroscopy tests (TGA/FT-IR) were employed to establish the impact of the of FRs on the flammability of the polymer.

## 2. Results and Discussion

### 2.1. Characteristics of Bio-Sourced Components

The share of cellulose and its crystallization degree determine the mechanical properties of plant fibers [17]. Moreover, hydrophilic cellulose and hemicellulose are responsible for maintaining the proper hydration of the fibers, while lignin gives plants stiffness and protects them against the action of microorganisms [18]. The high content of lignin and secondary components may weaken the thermal stability of the material [19]. It has also been proven that the share of individual ingredients may determine the burning behavior of polymers [11]. The share of plant fiber components and their elemental composition are listed in Table 1, while the SEM images of particles are shown in Figure 1.

The cellulose content in the plant components ranged from 30.7 to 33.4%. For comparison, the amount of this component in wood is 40–50%; in wheat husks, it is 23%, while in coconut and almond shells, it is 34% and 29%, respectively [20,21,22,23]. The share of hemicellulose was 14.3% for peanut shells and 21.6% for sunflower husks, which is similar to the data presented in the literature (rice husks, 14%; barley husks, 12%; coconut shells, 21%) [19,23]. Significant differences in the obtained values were also observed in the case of lignin; the share for SH was 23.1%, while for walnut shells, it was 30.2%. The content of this ingredient in plant raw materials is usually below 20% (cereal straw: 12- 17%, rye husks: 13%, wheat: 16%, jute fiber: 12%), and the noted values, especially for shells, are similar to cedar wood (34%) and coconut fiber (34–45%) [18,19,21,22,23,24]. The most significant difference in the chemical structure concerned fat, a higher amount of which was recorded for sunflower, a well-known oil plant. According to data in the literature, the fat in plants ranges from 0.2% for abaca to 4.2% for nettle [18]. The amount of water ranged from 6% (PS) to approximately 10% (SH).

The plant components were characterized by a similar elemental composition. A significant difference was noted only in the case of chlorine, which was three times higher for PS. Peanut shells also demonstrated a lower heat of combustion. The obtained values are similar to the heat of combustion of wood biomass [25].

The morphology of the plant components’ particles, as a result of the grinding process, was assessed based on SEM images (Figure 1). The particles within both fillers differed in size and shape. In the case of PS, a rough and developed surface, as well as a porous structure, can be observed. A porous and sometimes layered structure is also visible in the case of ground sunflower husks; however, the particles resulting from grinding peanut shells are more compact.

Figure 2 shows the particle size distribution function (Q3(x)) and its derivative (dQ3(x)) as a function of particle size (x). Despite employing the same grinding procedure, different particle sizes, depending on the type of bio-sourced component, were observed. The share of smaller particle sizes in the case of ground peanut shells was higher, and the graph shows one clear, although quite broad, peak. Meanwhile, for particles of ground sunflower husks, the range was even more extensive, and the curve shows two maxima, the first of which is similar to the PS, while the second gives the maximum value. The specific structure of the sunflower husk resulted in fiber disintegration in the longitudinal direction, causing the share of larger fiber sizes to be more significant [26].

### 2.2. Evaluation of the Properties and Structure of Michael Addition Polymers with Flame-Retardant Systems

#### 2.2.1. Chemical Structure

The FT-IR spectra of the obtained polymeric composites are shown in Figure 3. The conversion of the Michael addition polymer matrix can be judged by the lack of absorption bands at ~1630 cm^−1^ and ~810 cm^−1^, attributed to the absorbance of acrylic group vibrations, which were almost non-existent. This indicated that the content of free acrylic groups in polymers was low [27]. Thus, relatively high conversion occurred. The polymer matrix has both aromatic and aliphatic moieties in its chemical structure, and the aromatic moieties are identified as absorption bands at ~3000–2864 cm^−1^ as a C-H stretching vibration and at 730 cm^−1^ as out-of-plane C-H bending of an aromatic ring. The different flame-retardant additives, except for the APP + PS + MC sample, had little effect on the chemical structure of the polymer. A distinct N-H stretching vibration was identified at 3360 cm^−1^ and 3220 cm^−1^, amide stretching was identified at 1785 cm^−1^, C=N stretching vibration was identified at 1660 cm^−1^, and melamine sting stretching vibration was identified at 1530 cm^−1^, indicating the melamine cyanurate structure in the sample. The melamine cyanurate was the only additive that had a distinctive influence on the FT-IR spectra of the samples. The surface of the samples was analyzed using the FT-IR spectra collected in ATR mode. It was found that melamine cyanurate might have better solubility in the liquid components of the Michael thermoset resin and that the APP + PS + MC sample had distinctive absorption bands for the melamine cyanurate, while the other additives covered in the resin material did not yield signals in the ATR-FT-IR analysis. Moreover, the N-H_2_ groups of the melamine cyanurate could contribute to the formation of the polymer matrix via aza–Michael addition. Nevertheless, this does not hinder their ability to perform as flame retardants, especially where intumescent performance is required.

#### 2.2.2. Microstructure and Contact Angle

Microstructure observations were performed by SEM, and images of the fracture surface of the unmodified polymer and polymer with flame retardants are shown in Figure 4.

The SEM images revealed the presence of commercial IFR in the whole volume of materials. The occurrence of pores resulted from the loss of flame retardant particles, suggesting limited adhesion between the polymer and APP, PER, or M (Figure 4b). In the case of plant-origin components, the surface’s wettability by the polymer was excellent, and sometimes the phase boundary was difficult to mark. However, voids inside the particles of ground shells and husks (Figure 4g,h) may have weakened the material’s mechanical properties.

Figure 5 presents a cross-sectional image of the samples in their central part, obtained using 3D-CT, along with the porosity characteristics for each series. Meanwhile, Table 2 lists the calculated porosity values. The graphs on the right side of the 3D-CT images represent the porosity distribution, understood as pore volume versus pore diameter, in the analyzed part of the sample.

The analysis confirms the irregular distribution of flame retardant system particles throughout the entire volume of the investigated samples and the presence of agglomerates. The presented porosity values represent the total volume, including the porosities in the polymer and the polymer-filler interfacial region, as well as those contained within the filler itself. Therefore, an increased share of pores with a diameter of less than 10^−1^ mm is observed for all IFR-modified series, unlike in the reference sample (rP). This increased porosity, observed when using lignocellulosic fillers, was described in our earlier work relating to epoxy resins filled with fragmented wooden parts of *Pinus sibrica* [28]. Samples containing the flame-retardant system with SH revealed the highest porosity. This effect is related to both the bimodal distribution of particle sizes in SH, which leads to the increased viscosity of the composition and difficulties degassing it during forming [29], as well as the significantly higher porosity inside the particles compared to PS, which is visible in SEM images.

Contact angles were determined to exclude the negative impact of the bio-sourced component on the hydrophilicity of the polymer, which would favor the development of microorganisms on the surface and increase the degradation rate. If the contact angle is greater than 0° but less than 90°, the surface has hydrophilic properties, while values above 90° but below 150° are characteristic of a hydrophobic surface. The use of synthetic flame retardants contributed to an increase in the contact angle from 89 to 106° (Figure 6), which could be related to the increase in surface roughness due to the introduction of solid particles ranging from several to several dozen micrometers in size. In turn, the values for bio-sourced flame retardant systems range from 90 to 103°, meaning that introducing natural components together does not significantly affect the polymer’s hydrophilicity. Unfortunately, the multi-component systems made it difficult to observe the possible relationships between the share or type of plant component.

#### 2.2.3. Thermal, Mechanical and Thermomechanical Analyses

The heat-flow changes as a function of temperature obtained in the first DSC heating cycle are collectively presented in Figure 7. The glass transition temperatures determined from the DSC are summarized in Table 3. Apart from the phase transition interpreted as the glass transition temperature (T_g_), no additional significant thermal effects were noted that could indicate substantial changes in the polymer structure or temperature-induced reactions caused by the presence of flame-retardant systems. In the literature, the possibility of implementing the Michael reaction using pentaerythritol derivatives, such as pentaerythritol tetrakis(acetoacetate) or Michael acceptor dipentaerythritol pentaacrylate [30], as the Michael donor has been demonstrated. However, in the case of the FR systems investigated, this was unlikely to influence the course of the cross-linking process due to the lack of Michael’s active hydrogen atom. Regarding the possible impact of the modifiers used, only melamine derivatives can actively participate in polymer curing due to the presence of NH_2_ groups.

In order to determine the post-curing temperature of the composition, a multi-heating test by DSC was carried out. Selected rP and APP + PER + M samples were subjected to repeated annealing at increasing maximum cycle temperatures (Figure 8a). Based on the recorded heat flow runs (Figure 8b), it can be concluded that the glass transition temperature did not change significantly, and the use of a 70 °C post-curing temperature was sufficient to obtain cast samples with a stable structure.

Figure 9 shows the thermomechanical curves, with changes in the storage modulus (G’) and damping factor (tanδ) as a function of temperature, obtained from DMA (in both graphs, the post-curing temperature (70 °C) was additionally marked). Introducing FRs increased the modified materials’ storage modulus in the temperature range below and above the glass transition temperature (−20 °C and 80 °C). This beneficial effect, connected with the improvement of sample stiffness, is related to the reinforcing effect of dispersed rigid additives structure, as it was noticed in our previous studies for unsaturated polyester resin modified with halogen-free flame retardants [31]. A slight increase in the storage modulus above relaxation at 0–60 °C is observed for all materials (Table 3). Due to the lack of additional effects in the temperature range above the glass transition temperature on the DSC curve, it can be assumed that the additives did not affect the temperature-induced post-curing processes of the polymer. Introducing flame retardants decreased the T_g_ values determined by the DSC and DMA methods (Table 3), and the most significant changes were noted in the case of systems containing melamine derivatives, which may be related to the potential reactivity of melamine derivatives in Michael addition, which are involved in aza–Michael reactions with Michael acceptors [32]. In conclusion, the understanding of the effects of FRs on changes in thermal and thermomechanical properties should be limited to the reduced miscibility of the system and partially hindering the creation of a hyper-branched or cross-linked structure of the matrix domain instead of reactivity of the components with matrix (excluding melamine).

Figure 10 summarizes the results of mechanical tests (impact strength, hardness) and the Vicat softening point (VST). The addition of flame-retardant systems resulted in a significant reduction in the impact strength of the materials compared to the unmodified matrix. This effect appears as a result of the introduction of insoluble domains into polymers, which is characteristic of composites with particles and materials with a flame retardant [33,34]. The presence of additives constituting a separate dispersed phase with limited adhesion leads to the formation of micro-notches during dynamic loading, forming additional crack propagation start points and significantly reducing the impact strength [35]. All tested materials containing additives showed comparable impact strength values, which results from the same share used and similar geometrical characteristics of the additives.

The addition of FRs caused a decrease in the hardness compared to rP, which was determined by the Shore D method. The highest values among the modified materials were recorded for the sample containing APP + PER + M and MPP + PS, which was characterized by the smallest number of pores. The decrease in hardness can be correlated with changes in the polymer cross-linking densities, which, considering the differences in T_g_ observed by the DSC and DMA methods, suggests the correctness of this statement [36]. However, it should be emphasized that all samples showed an acceptable hardness, with the lowest value being 57°ShD for the MP + PS. All modified samples showed an increased VST value compared to unmodified rP, with the most beneficial effect observed for APP + PER + M.

### 2.3. Assessment of the Effectiveness of the Developed Systems on the Michael Reaction Polymer Flammability

#### 2.3.1. Thermal Stability

The results of the TG analysis are summarized in Table 4 and shown in Figure 11, where the curves of mass loss and their derivatives are displayed as a function of temperature. In addition to the analysis carried out in an inert atmosphere, measurements in the air, to determine the yield of residue in the presence of oxygen, were also made.

The 5% mass loss (T_5%_) for the polymer, corresponding to the onset of thermal decomposition, was 324 °C, while for the other samples, a significant reduction from 19 to 61 °C was noticed. Decomposition at lower temperatures, attributed to char formation, is characteristic of intumescent flame retardants [37]. A slight peak in the DTG curve for rP was found around 360 °C; however, the most intense degradation, related to the decomposition of the polymer, appeared above 400 °C. In the case of the polymer modified with flame-retardant systems, additional steps corresponding to the sublimation of melamine (DTG1) and the dehydration of phosphoric acid (DTG2) [38] can be observed (Table 4). Melamine sublimes at about 350 °C, but volatilization starts at a lower temperature, which allows for the continuous formation of the swollen layer [4]. The decay of bio-based components, such as hemicellulose, lignin, and cellulose, which degrade at temperatures of 270–300 °C, 330–370 °C and 190–450 °C, respectively, overlaps with the decomposition of the polymer and FRs, making them difficult to determine [11]. The stages above 600 °C correspond to the decomposition of transient char (Figure 11) [39].

Using multi-component FR systems led to a shift in peaks towards lower temperatures; however, the lower decomposition rates led to a much higher residue yield. The amount of residue in nitrogen was more than 2.5 times greater than that for the polymer (APP + SH, APP + PS + MC), while in the air, it increased from 0% (rP) to a maximum of almost 9% (APP + PS + SH + M). Enhancing the formation of char is advantageous from a flame hazard point of view. The lack of a linear relationship can be attributed to the multi-component composition of systems and the small size of the samples employed for the analysis.

#### 2.3.2. Flammability

Pyrolysis combustion flow calorimetry (PCFC) allows the milligram-sized samples’ flammability to be evaluated by reproducing the processes that occur during burning without maintaining a flame [40]. The course of the samples’ heat release rate (HRR) is shown in Figure 12, while Table 5 gives the combustion process parameters.

Apart from rP, the thermal degradation occurred in two stages (Figure 12), with heat release in each [41]. For all the samples investigated, except APP + PS + SH + M, a decrease in pHRR compared to the unmodified polymer was achieved, and the lowest value was determined for MP + PS (reduction by 45%). Replacing PER with two plant components that significantly differed in grain size affected the distribution of FR components in APP + PS + SH + M. In most cases, multi-component systems were less effective, especially in systems with larger SH particles, probably due to the samples’ less homogeneous structure. Similar to TGA, the introduction of FRs shifted the temperature of pHRR (T_pHRR_) towards lower temperatures (excluding again APP + PS + SH + M). The lowest total heat release (THR) was recorded for APP + PS + MC (15.5 kJ/g); however, a reduction was observed for all materials with FRs due to the replacement of some parts of the polymer with less flammable components. The heat release capacity is a significant indicator used to assess the flame hazard of materials [40]. The HRC of rP was 252 J/g·K, and despite the reduction for most samples, only for MP + PS was the obtained value below 200 J/g·K, reaching 152 J/g·K.

Since cellulose may have a similar effect to PER in IFRs, the plant component has been used as a carbon source in the flame retardant systems developed, but also partly as a diluting and swelling factor, resulting from the decomposition of hemicellulose and the release of CO_2_ and H_2_O. In most cases, replacing the synthetic carbon source with a lignocellulosic component in APP + PER + M brought a negative effect, and only for APP + PS + MC was a value close to the commercial IFR achieved. Substituting pentaerythritol with ground peanut shells was effective in this system and allowed the bio-source FRs to be obtained. The presence of phosphorus led to the phosphorylate reaction, while melamine salt (MC) favored the progressive condensation of melamine instead of volatilization. In systems where APP was replaced by MPP, the maximum heat release rate was similar, with the type of plant component used being of secondary importance. In turn, replacing the MPP with MP, characterized by a higher nitrogen content, resulted in a significant reduction in pHRR. In conclusion, this elevated amount of nitrogen is a critical factor, whichaccelerates the phosphorylation of cellulose by forming a P–N intermediate and allows for the effective retention of decomposition products in the condensed phase.

#### 2.3.3. Evolved Gas Analysis

An evolved gas analysis of the thermal decomposition product was performed via a TGA/FT-IR experiment, where the released products were flushed through the FT-IR gas cell, and the IR absorption data was collected. Three-dimensional images of the evolved gas FT-IR spectra and the FT-IR spectra during thermal degradation are depicted in Figure 13. The neat polymer sample released products into the gaseous phase at higher temperatures than the samples modified with FR systems. The polymer matrix decomposed into H_2_O, CO_2_, and CO products that were detected at ~3670 cm^−1^, ~2310 cm^−1^, and ~669 cm^−1^ and 2100 cm^−1^, respectively (Figure 13a). The volatile organic compounds of the polymer matrix decomposition were identified at ~2970 cm^−1^ as a C-H stretching vibration and at 730 cm^−1^ as the out-of-plane C-H bending of an aromatic ring. Moreover, the polymer matrix degraded into esters and acids, as the carboxylic group C=O stretching vibration was identified at ~1770 cm^−1^. The intensity of the absorption bands, corresponding to organic compounds, was more intensive for products released in the first stages of the matrix decomposition at ~360 °C, whereas, at later decomposition stages, H_2_O, CO_2,_ and CO were released. The APP+PER+M and MP+PS samples had products of sublimation of melamine, and dehydration of phosphoric acid were identified at ~1550 cm^−1^ and 1120 cm^−1^, respectively [42,43]. Both polymers with bio-based FR systems released products into the gaseous phase sooner than rP; however, no highly toxic gasses, such as HCN, were identified, which could indicate that the selected FR systems could be upscaled for further fire retardancy material development [12,28].

#### 2.3.4. The Mechanism of Flame Retardancy of Developed FR System for Michael Addition Polymer

Due to the presence of polar groups such as carboxylic acids and high concentrations of hydroxyl groups, the structure of lignocellulose components is appropriate for improving the effectiveness of phosphorus flame retardants. The plant flame retardancy potential is due to the chemical structure and the possible interaction with decomposition products of polymer containing IFR, which are susceptible to phosphorylation. Plenty of reactive areas lead to the retention of P-rich polyaromatic substances in the condensed phase. This reaction increases residue retention, where phosphorylation via transesterification occurs while the polymer undergoes dehydration and cyclization reactions [4,5,44]. Moreover, the presence of phosphorus and nitrogen in the flame-retardant system causes the formation of compounds, such as phosphorus oxynitrides and phosphazenes, which increases the yield and stability of residues [5,45,46].

The presented research results confirm that lignocellulosic raw materials can be a carbon source and successfully replace the ingredients of intumescent flame retardants or enhance their effectiveness. A critical factor in developing the system is the chemical composition of plant components. A more favorable effect was observed for ground peanut shells with a grain size similar to synthetic FR, containing more aromatic lignin and cellulose, which provides hydroxyl groups for the esterification reaction, instead of the less thermally stable hemicellulose. In turn, in the case of synthetic ingredients, the nitrogen content is crucial.

## 3. Experimental

### 3.1. Materials and Preparation

As component A, aromatic polyester polyol Neopolyol 380 from NEO GROUP (Vilnius, Lithuania), obtained from industrial PET waste and acetylated with tertbutyl acetoacetate in the transesterification reaction, was employed. The obtained NEO380_AA was characterized by an acid number of <5 mg KOH/g, a hydroxyl number of 40.7 mg KOH/g, a water content of less than 0.05%, and an acetoacetate group content of 0.4242 mol·100 g^−1^. Component B was trimethylpropane triacrylate (TMPTA), a trifunctional acrylic ester monomer obtained from trimethylolpropane and manufactured by MERCK. 1,1,3,3-tetramethylguanidine (TMG) from MERCK (Darmstadt, Germany) was used as a catalyst. The composition ingredients were proportioned by weight: 100 g of component A, 83 g of component B and 0.74 g of the catalyst. The pot life of the composition was ca. 6 min.

As the component of the flame retardant system, 6 commercial FRs were selected, as listed in Table 6.

Sunflower husks (*Helianthus annuus* L. *species*, SH) and peanut shells (*Arachis hypogaea* L. *species*, PS) were provided by local suppliers (Poland, Central Europe). The drying process was performed using a laboratory dryer for at least 3 days at 65 ± 5 °C, while grinding was conducted on a laboratory sieve mill L-0210 from Cloer (Arnsberg, Germany).

FRs were introduced to component A and mixed using a mechanical stirrer with a rotational speed of 1000 for 10 ± 0.5 min.; then, the mixture was degassed in a vacuum dryer (approx. −1 MPa) for 10 ± 0.5 min. Simultaneously, a catalyst was added to component B, stirred at a rotational speed of 500 for about 1 min and degassed for 5 ± 0.5 min. Then, both mixtures were mixed up and stirred again for about 30 s at a rate of 500 rpm. After degassing for 1 min, the mixture was poured into open molds. The samples were cured at room temperature for 24 ± 2 h and at 70 °C for 3.5 ± 0.5 h.

The total share of FR systems in the polymer was 20 wt.%. The amount of each FR was based on the most common intumescent flame retardant (IFR), in which the proportion of APP: PER: M was 3:1:1. Table 7 lists the devised formulations of bio-based FR systems, along with reference materials in the form of an unmodified polymer and polymer (rP) with an intumescent system (APP + PER + M).

### 3.2. Methods

Laboratory analyses of the basic thermophysical parameters and chemical composition of the tested plant material were carried out in the laboratories of the Department of Genetics, Plant Breeding and Bioresources Engineering at the University of Warmia and Mazury in Olsztyn, in accordance with the standard methodology used in these laboratories [47,48]. The content of substances soluble in cold (20–25 °C, 48 h) and hot (100 °C, 3 h) water was defined using the ANKOM A200 apparatus (Ankom Technology, Macedo, NY, USA). The [49] standard determined the fractions of neutral detergent fiber in biomass, while acid detergent fiber and acid detergent lignin followed the [50] standard. These analyses allowed for the determination of the content of lignin, cellulose and hemicellulose in the samples, in accordance with the application note “FOSS AN 304”. The calculations used were consistent with the detailed description presented in the work of Stolarski et al. [47]. All laboratory analyses were performed in triplicate for each type of tested plant material.

The moisture content was determined by weighing and drying at 105 °C [51] using an FD BINDER laboratory dryer from Binder GmbH (Tuttlingen,, Germany). The heating value (HV) was defined using the dynamic method in an IKA C2000 calorimeter (IKA^®^-Werke GmbH & Co, Taufen, Germany). The fat content was determined based on the [52] standard using the Soxhlet extraction method, employing the Büchi B-811 automatic extraction system (Büchi Labortechnic AG, Flawil, Switzerland) and petroleum ether with a boiling point of 40–60 °C. The content of chemical elements such as carbon (C), hydrogen (H) and sulfur (S) was determined using the ELTRA CHS 500 automatic analyzer (ELTRA GmbH, Neuss, Germany) according to the [53,54] standards. In turn, the nitrogen (N) content was determined using the Kjeldahl method utilizing the K-435 mineralizer and the BUCHI B-324 distiller (Büchi Labortechnic AG, Flawil, Switzerland). The chlorine (Cl) content was determined following the [55] standard, which includes, among others, firing samples of plant material in a NABERTHERM muffle furnace (Nabertherm GmbH, Lilienthal, Germany) at a temperature of 650 °C with the addition of the Eschka mixture, the preparation of aqueous extracts, and titration with 0.025 mol/L AgNO_3_. The composition of individual chemical elements in the plant material is given in % DM.

The appearance of particles of raw plant materials after the grinding process and the microstructure of the samples were assessed using a scanning electron microscope TM3000 (Hitachi, Tokyo, Japan) equipped with a backscattered electron detector (BSE). The samples were fixed to carbon tape and coated with gold using POLARON SC7640 (Quorum Technologies Ltd., Newhaven, UK). Images were taken with an accelerating voltage of 5–15 kV. A magnification of 250× was used.

The particle size distribution of the fillers was analyzed using the laser particle sizer apparatus ANALYSETTE 22 (Fritsch GmbH, Weimar, Germany), operated in the 0.08–2000 μm range. The cumulative size distribution Q3(x) and adequate histogram dQ3(x) were considered during the analysis. The analysis was performed in three replicates.

Fourier transform infrared spectroscopy was performed using an FT-IR Nicolet 6700 spectrometer (Thermo Electron Corporation, Waltham, MA, USA) in the Attenuated Total Reflectance (ATR-FT-IR) mode, using 64 counts. Moreover, the components were thoroughly mixed with KBr and pressed into pellets. The analyses were carried out using Omnic 8 software in the spectral range 400–4000 cm^−1^ and with a resolution of 4 cm^−1^.

The structure of the samples was examined using X-ray tomography, using the v|tome|x s240 model (Waygate Technologies/GE Sensing & Inspection Technologies GmbH, Hürth, Germany). The use of micro-computed tomography (μCT) was focused on evaluating the distribution of the pores and the additives/filler distribution. After scanning, the following parameters were used during the measurements: a microfocus X-ray tube (voltage 150 kV/current 200 μA); an exposure time for one picture of 150 ms; and a voxel size of 15 μm.

The wettability of the surfaces was assessed by measuring the water contact angles. The measurements were performed using an OCA15 goniometer (DataPhysics Instruments, Filderstadt, Germany) with the SCA20 6.1 software. The contact angle was determined using the needle-in method at room temperature, while the volume of the water droplets was 5 μL. The values were the average of ten measuring points on the sample’s surfaces.

Differential scanning calorimetry measurements were performed using a Netzsch DSC 204F1 Phoenix apparatus (Netzsch GmbH, Selb, Germany). Material samples of 10 ± 0.05 mg were placed in aluminum crucibles with pierced lids and then heated and cooled within the range of −50 °C to 180 °C at a heating/cooling rate of 10 °C/min in a nitrogen atmosphere.

The dynamic–mechanical properties of the specimens with dimensions of 50 × 10 × 4 mm were measured using an MCR 301 (Anton Paar GmbH, Graz, Austria) in torsion mode, operating at a frequency of 1 Hz in the temperature range of −130 °C and 180 °C and at a heating rate of 3 °C/min. The position of tanδ at its maximum was used to determine the glass transition temperature (Tg).

The impact strengths of the 10 × 4 × 15 mm unnotched samples were measured by the Dynstat method [56] on 4 × 10 × 15 mm samples. A Dys-e 8421 apparatus (Leipzig, Germany) equipped with a 0.98 J hammer was employed.

A hardness evaluation was conducted using a durometer HBD 100-0 Shore D (Sauter GmbH, Wutöschingen, Germany) according to the [57].

Vicat softening point temperature (VST) investigations were performed using a CEAST HV3 apparatus (Instron, Norwood, MA, USA). The measurements were carried out in an oil bath following the [58] standard. The Vicat softening temperature was determined in the B50 measurement configuration, i.e., a load of 50 N and a heating rate of 50 °C/h. The experiments were conducted for six specimens from each series.

The thermal decomposition was examined in nitrogen and air by employing the thermal analyzer TGA Q500 (TA Instruments Ltd., Mildford, CT, USA). The 10 mg samples were heated at 10 °C/min from room temperature to 1000 °C. The gas flow rate was 10 mL/min in the chamber and 90 mL/min in the furnace.

An analysis of flammability was carried out using a Pyrolysis Combustion Flow Calorimeter (Fire Testing Technology Ltd., East Grinstead, UK) following ASTM D7309 [59]. The decomposition was carried out in an inert gas atmosphere in the 150–750 °C temperature range with a heating rate of 1 °C/s. After pyrolysis, the gas stream was mixed with oxygen and entered into a combustor at 1000 °C. The oxygen concentrations and flow rates of the combustion gases were used to determine the oxygen depletion involved in the combustion process, the heat release, and the heat release capacity. The mass of the tested samples was 3 ± 0.7 mg.

The Fourier transform infrared Nicolet 6700 spectrometer (Thermo Electron Corporation, Waltham, MA, USA) was connected with a TGA Q500 (TGA/FT-IR) thermal analyzer to investigate the formation of volatiles produced in the thermal decomposition process. The analyses were performed in the spectral range of 400–4000 cm^−1^ and at a resolution of 4 cm^−1^. Then, 10 ± 0.5 mg samples were heated in the air from above 30 to 1000 °C at 10 °C/min. The FT-IR gas cell was held at 240 °C, while the transfer line temperature was set to 250 °C.

## 4. Conclusions

This work provides a comparative investigation of the impact of flame-retardant systems based on plant components on the properties of a new Michael addition polymer. The subject of the present study was a composition consisting of intumescent flame retardants (APP + PER + M), in which lignocellulose ones, such as sunflower husks and peanut shells, replaced a part of the synthetic components. The analysis was extended to include other popular IFRs, such as MPP and MP, for which flame-retardant systems were also developed.

It was demonstrated that the investigated systems did not adversely affect the cross-linking process, but there were some changes in the polymer cross-linking density, assessed by DSC and DMA. Moreover, the new FRs resulted in a decrease in the hardness and impact strength, as well as an increase in the Vicat softening point and contact angle values.

Pyrolysis combustion flow calorimetry proved that MP + S reduces the heat release rate and heat release capacity most effectively. The flame retardant system led to a decrease in the mass loss rate and an increase in the char yield, as suggested by TG analysis. The investigation showed that an elevated amount of nitrogen in MP accelerates the phosphorylation of cellulose and allows for the effective retention of decomposition products in the condensed phase. The values of the mentioned indicators for MP + PS were significantly lower both for the unmodified polymer and the polymer with APP + PER + M. The results confirm that lignocellulosic raw materials can be a carbon source and replace the components of intumescent flame retardants.

## Figures and Tables

**Figure 1 molecules-30-02556-f001:**
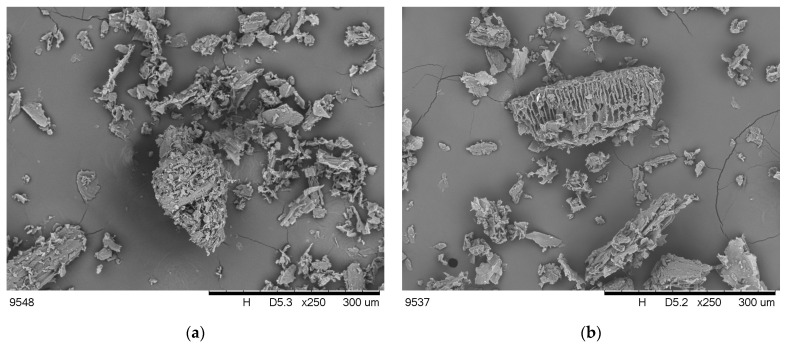
SEM images of SH (**a**) and PS (**b**) components.

**Figure 2 molecules-30-02556-f002:**
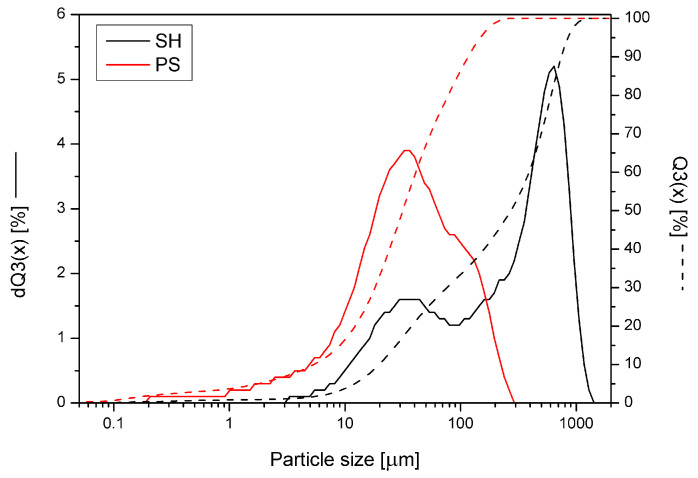
Representative particle size distribution curves of bio-source components.

**Figure 3 molecules-30-02556-f003:**
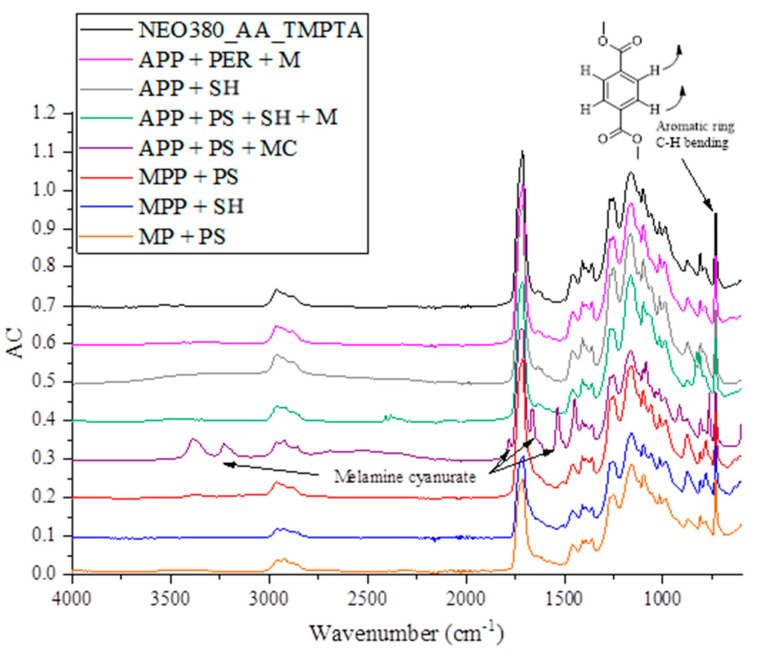
FT-IR spectrum of investigated samples.

**Figure 4 molecules-30-02556-f004:**
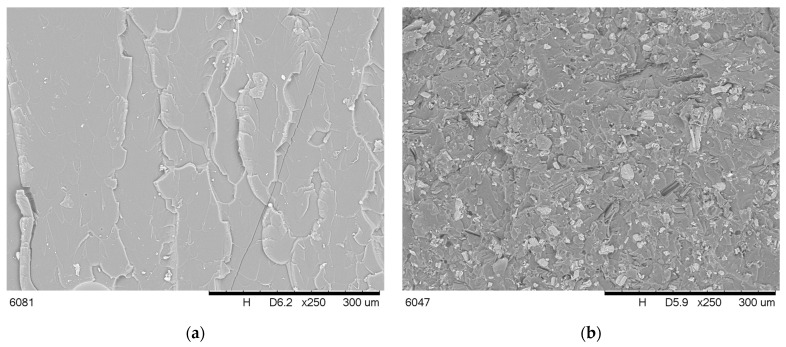
SEM images of rP (**a**), APP + PER + M (**b**), APP + SH (**c**), APP + PS + SH + M (**d**), APP + PS + MC (**e**), MPP + PS (**f**), MPP + SH (**g**) and MP + PS (**h**).

**Figure 5 molecules-30-02556-f005:**
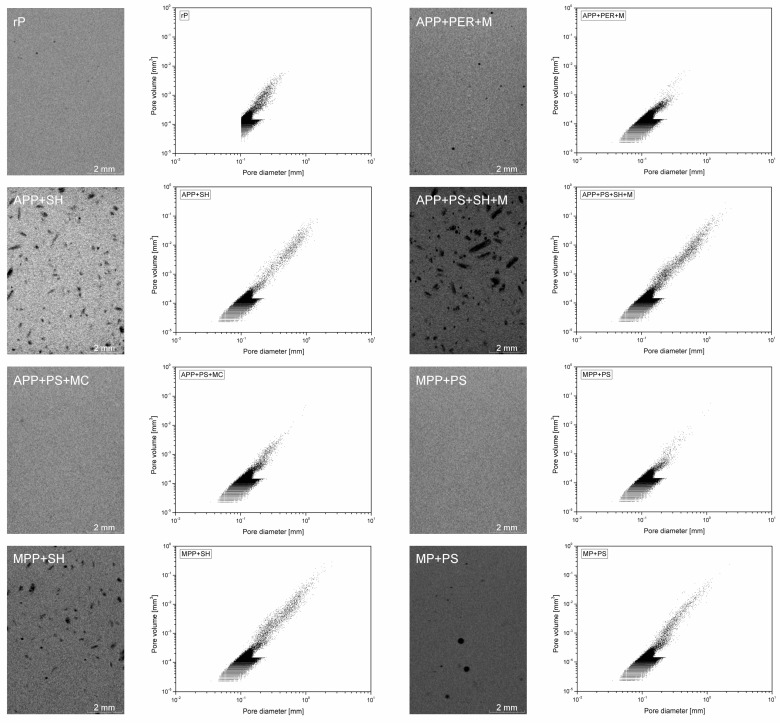
Three-dimensional tomography image of the center part of the specimens and porosity distribution in samples (pore volume vs. diameter plots).

**Figure 6 molecules-30-02556-f006:**
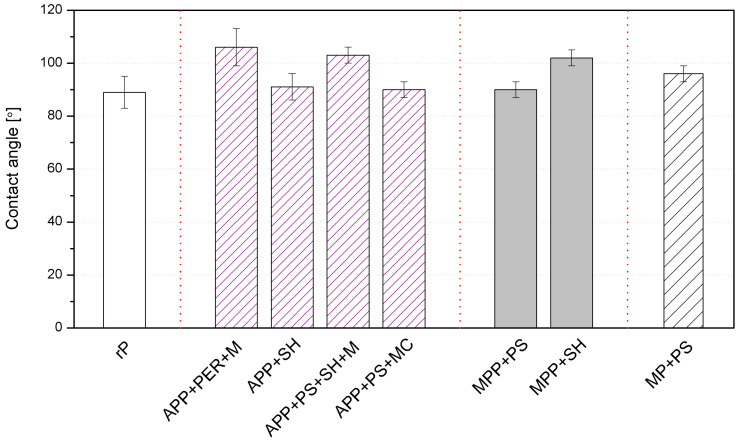
Contact angles of the Michael addition polymer with FR systems.

**Figure 7 molecules-30-02556-f007:**
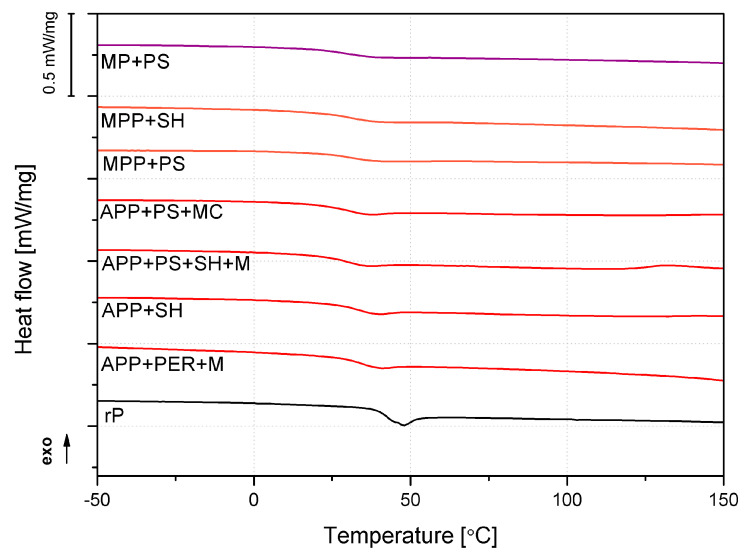
DSC first heating thermograms of the unmodified and flame-retardet polymer.

**Figure 8 molecules-30-02556-f008:**
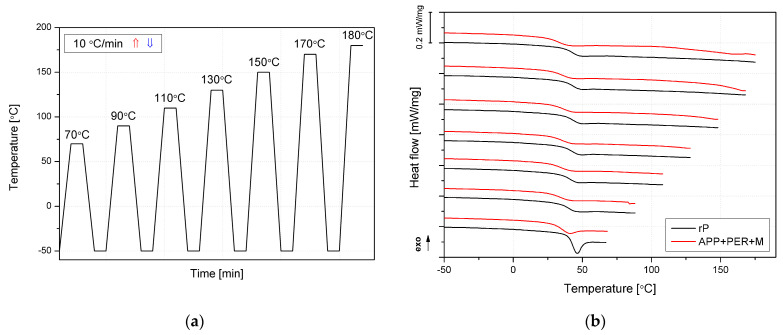
Differential scanning calorimetry multi-heating experiment temperature program (**a**); DSC curves for rP and APP + PER + M (**b**).

**Figure 9 molecules-30-02556-f009:**
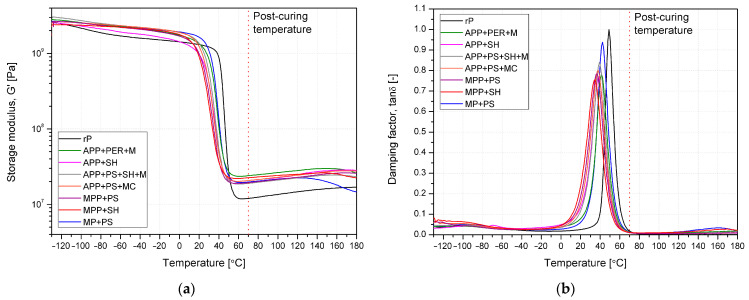
DMA results for unmodified and modified matrix; storage modulus (**a**) and damping factor (**b**) vs. temperature curves.

**Figure 10 molecules-30-02556-f010:**
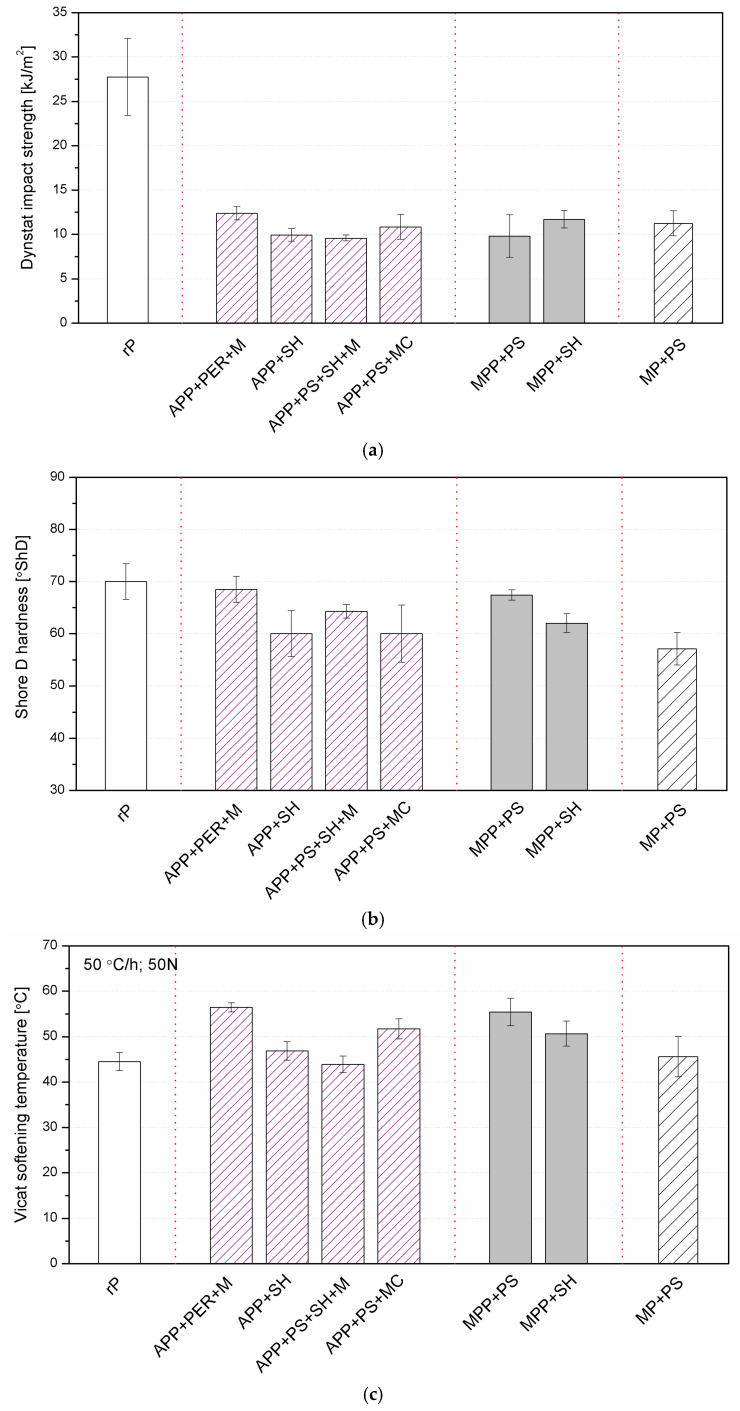
Results of mechanical and thermomechanical properties: Dynstat impact strength (**a**); Shore D hardness (**b**); Vicat softening temperature (**c**).

**Figure 11 molecules-30-02556-f011:**
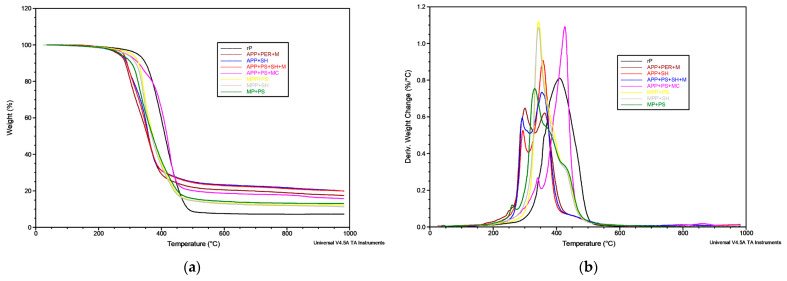
(**a**) Mass loss and (**b**) derivative of mass loss as functions of temperature from the thermogravimetric analysis.

**Figure 12 molecules-30-02556-f012:**
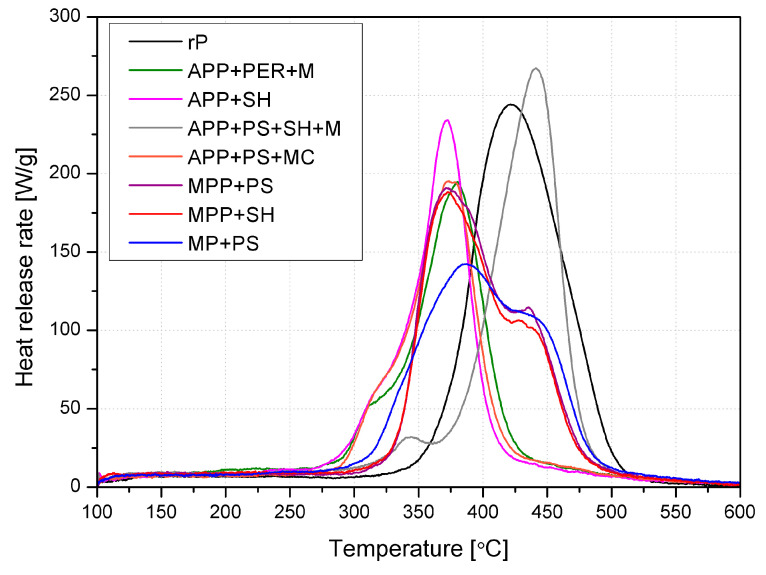
Representative heat release rate curves as temperature functions from PCFC analysis.

**Figure 13 molecules-30-02556-f013:**
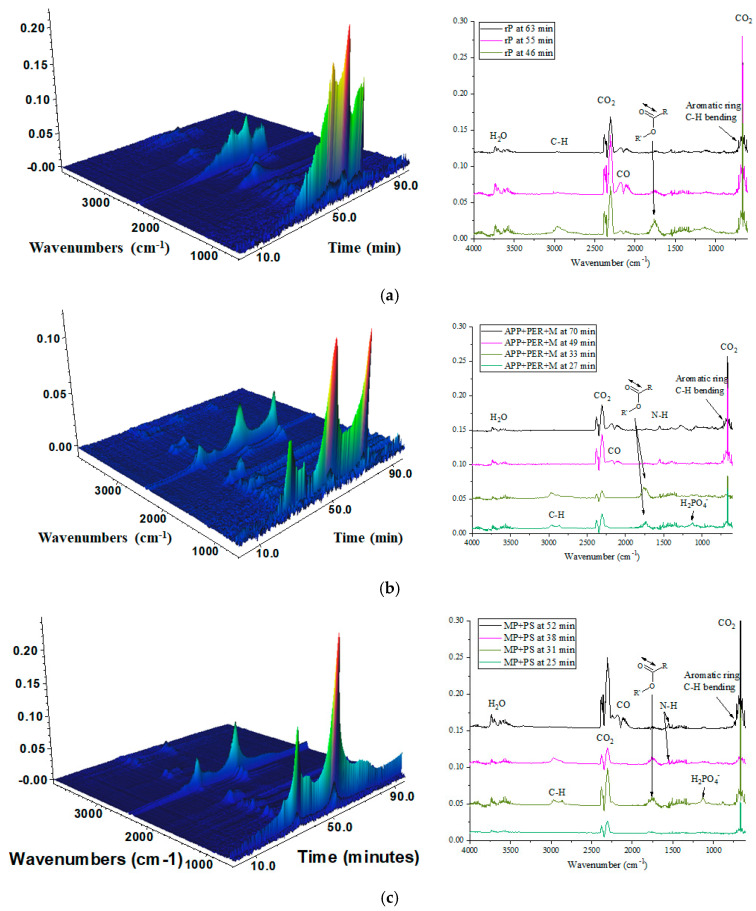
Evolved gas analysis from TGA/FT-IR spectra for (**a**) rP, (**b**) APP + PER + M and (**c**) MP + PS polymer composite samples.

**Table 1 molecules-30-02556-t001:** The chemical composition and heat of combustion bio-sourced components.

Bio-Sourced Component	Celulose [%]	Hemicellulose [%]	Lignin [%]	Fat [%]	Humidity [%]	C [%]	N [%]	H [%]	S [%]	Cl [%]	Heat of Combustion [J/g]
SH	30.7 ± 0.2	21.6 ± 0.2	23.1 ± 0.1	2.7 ± 0.1	9.8 ± 0.3	47.7 ± 0.1	0.9 ± 0.0	5.6 ± 0.1	0.1 ± 0.0	0.1 ± 0.0	20,081 ± 46
PS	33.4 ± 0.4	14.3 ± 0.3	30.2 ± 0.3	1.4 ± 0.0	6.1 ± 0.1	47.4 ± 0.2	1.0 ± 0.0	6.0 ± 0.1	0.1 ± 0.0	0.3 ± 0.0	19,570 ± 30

**Table 2 molecules-30-02556-t002:** Porosity of investigated samples.

Material	Porosity [%]
rP	3.48
APP + PER + M	3.99
APP + SH	7.78
APP + PS + SH + M	8.05
APP + PS + MC	3.44
MPP + PS	2.89
MPP + SH	6.91
MP + PS	4.67

**Table 3 molecules-30-02556-t003:** Thermal parameters obtained by DSC and DMA.

Material	Tg DSC	Tg DMA	tanδ at Tg DMA	G’ at −20 °C	G’ at 20 °C	G’ at 80 °C
	[°C]		[-]	[Pa]		
rP	40.9	49.0	1.003	1.51 × 10^9^	1.31 × 10^9^	1.27 × 10^7^
APP + PER + M	31.6	42.4	0.938	2.05 × 10^9^	1.63 × 10^9^	2.04 × 10^7^
APP + SH	32.0	41.3	0.782	1.96 × 10^9^	1.34 × 10^9^	2.47 × 10^7^
APP + PS + SH + M	29.7	37.9	0.769	1.64 × 10^9^	9.61 × 10^8^	2.13 × 10^7^
APP + PS + MC	28.7	39.1	0.839	2.08 × 10^9^	1.34 × 10^9^	1.96 × 10^7^
MPP + PS	29.4	38.0	0.801	2.09 × 10^9^	1.22 × 10^9^	2.13 × 10^7^
MPP + SH	28.1	37.0	0.785	1.92 × 10^9^	9.79 × 10^8^	2.00 × 10^7^
MP + PS	26.7	34.7	0.755	2.01 × 10^9^	8.79 × 10^8^	2.31 × 10^7^

**Table 4 molecules-30-02556-t004:** The results of the TG analysis.

Material	T_5%_ [°C]	DTG1 [°C; %/min]	DTG2 [°C; %/min]	DTG3 [°C; %/min]	DTG4 [°C; %/min]	Residue at 1000 °C in Inert Atmosphere	Residue at 1000 °C in Air
rP	324	-	-	363; 0.50	410; 0.81	7.32	0.00
APP + PER + M	263	251; 0.09	300; 0.65	363; 0.62	453; 0.06	17.42	1.73
APP + SH	270	-	294; 0.52	358; 0.90	440; 0.07	19.85	2.15
APP + PS + SH + M	289	242; 0.03	295; 0.01	342; 0.27	426; 1.09	15.71	8.78
APP + PS + MC	279	241; 0.03	291; 0.59	354; 0.73	452; 0.06	19.88	1.91
MPP + PS	305	273; 0.04	-	343; 1.12	425; 0.34	12.09	3.71
MPP + SH	287	242; 0.03	-	342; 1.02	423; 0.33	11.28	0.99
MP + PS	268	260; 0.12	-	332; 0.75 363; 0.55	423; 0.34	13.03	2.73

**Table 5 molecules-30-02556-t005:** The results of the PCFC analysis.

Material	pHRR [W/g]	T_pHRR_ [°C]	THR [kJ/g]	HRC [J/g·K]
rP	252 ± 16	417 ± 3	22.6 ± 0.3	277 ± 23
APP + PER + M	192 ± 10	377 ± 3	16.1 ± 0.5	211 ± 9
APP + SH	236 ± 4	370 ± 2	16.3 ± 0.2	260 ± 7
APP + PS + SH + M	268 ± 18	442 ± 2	19.9 ± 0.5	294 ± 19
APP + PS + MC	193 ± 11	372 ± 2	15.5 ± 0.5	210 ± 12
MPP + PS	191 ± 6	372 ± 3	19.5 ± 0.5	209 ± 6
MPP + SH	186 ± 7	373 ± 3	19.5 ± 0.4	204 ± 10
MP + PS	138 ± 6	384 ± 4	18.2 ± 0.8	152 ± 5

**Table 6 molecules-30-02556-t006:** The trade name, properties and supplier of the flame retardants used.

	Ammonium Polyphosphate (APP)	Dipentaerythritol (PER)	Melamine (M)	Melamine Cyanurate (MC)	Melamine Polyphosphate (MPP)	Melamine Phosphate (MP)
Trade name and supplier	Addforce FR APP201, Walter Thieme Handel GmbH (Stade, Germany)	Addforce^®^ FR Penta D40, Walter Thieme Handel GmbH (Stade, Germany)	Addforce FR FA405, Walter Thieme Handel GmbH (Stade, Germany)	Exflam MCA25, Grolman Sp. z o.o. (Neuss, Germany)	Exflam MPP, Grolman Sp. z o.o. (Neuss, Germany)	Exflam MP, Grolman Sp. z o.o. (Neuss, Germany)
Particle size	D_98_ < 15 μm	D_98_ = 40 μm	D_98_ = 40 μm	D_98_ < 25 μm	D_98_ < 8 μm	D_50_ < 10 μm
P and N content or another main component [%]	P min. 31% N min. 14%	C_10_H_22_O_7_ 85% (grupy hydroksylowe 38–40%)	C_3_H_6_N_6_ 99.8%	C_6_H_9_N_9_O_3_ min. 99.5%	P 42–44% N 12–14%	P min. 12% N 36–38%

**Table 7 molecules-30-02556-t007:** The formulations of bio-sourced flame-retardant systems.

Designation	APP	PER	M	MC	MPP	MP	SH	PS
rP								
APP + PER + M	3	1	1					
APP + SH	3						2	
APP + PS + SH + M	3		1				0.5	0.5
APP + PS + MC	3			1				1
MPP + PS					3			2
MPP + SH					3		2	
MP + PS						3		2

## Data Availability

Data is available upon request from the authors.

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
