# Peer review of "Bio-Based Flame-Retardant Systems for Polymers Obtained via Michael 1,4-Addition"

_molecules, 2025, doi:10.3390/molecules30122556_

Round 1

Reviewer 1 Report

Comments and Suggestions for Authors

The current paper assessed the flammability of a new polymer synthesized by Michael 1, 4-addition (rP) modified with developed intumescent fire retardant systems (FRs) in which lignocellulose components, such as sunflower husk (SH) and peanut shells (PS), replaced a part of the synthetic ones. The thermal, thermodynamical and fire retardant properties of the rP with 20 wt% of each from six FRs were investigated. However, there are several issues that should be addressed before publication. The specific comments are given below:

  1. In the introduction, please introduce the background why selected polymers obtained via 2 Michael 1,4-addition as the matrix.
  2. In Fig. 3, what is the meaning of “S”?
  3. “In turn, the values for bio-sourced fire retardant systems range from 90 to 103°, meaning that introducing natural and synthetic components together prevents increasing the polymers hydrophilic” (Lines 230-232). These statements are not true. Actually, the addition of MPP+PS did not change the contact angles of the Michael addition polymer. Please revise it.
  4. The APP+PS+SH+M showed higher pHRR and HRC than rP. Please comment it.
  5. The MP+PS showed the lowest pHRR and HRC. Please discuss the mechanism.
  6. “dehydration of phosphoric acid 400 were identified at ~1550 cm-1 and 1120 cm-1, respectively”. Please provide the references to support it.
  7. In addition to the Pyrolysis Combustion Flow Calorimeter, the LOI and UL-94 vertical burning tests are more essential to assess the fire retardant properties.

Author Response

The answer to Reviewer 1:

We would like to thank the Reviewer for all his valuable remarks. We agree with the recommendations that the manuscript should be improved and so an effort has been made to correct the article according to the comments.

  1. In the introduction, please introduce the background why selected polymers obtained via 2 Michael 1,4-addition as the matrix.

Suitable corrections according to Reviewer’s comment have been introduced in the introduction section.

  1. In Fig. 3, what is the meaning of “S”?

The authors wish to thank the Reviewer for drawing attention to the error that was made.

  1. “In turn, the values for bio-sourced fire retardant systems range from 90 to 103°, meaning that introducing natural and synthetic components together prevents increasing the polymers hydrophilic” (Lines 230-232). These statements are not true. Actually, the addition of MPP+PS did not change the contact angles of the Michael addition polymer. Please revise it.

We appreciate the Reviewer's remarks and agree that the addition of MPP+PS did not change the contact angles of the Michael addition polymer. The wording has been improved by changing "together prevents increasing the polymer's hydrophilicity" to "does not significantly affect the polymer's hydrophilicity".

  1. The APP+PS+SH+M showed higher pHRR and HRC than rP. Please comment it.

The use of the APP+PS+SH+M system increased pHRR and HRC compared to rP, giving the opposite effect to the intended one. However, the values ​​are within the standard deviation, so the flammability increase should be considered small. Replacing PER with 2 plant components, which significantly differ in grain size, affected the distribution of FR components in the polymer and caused an increase in porosity. As a result, the use of multi-component systems gave a worse effect compared to systems with a similar composition but a smaller share of different components. We thank the Reviewer for the remarks, the article has been modified accordingly.

  1. The MP+PS showed the lowest pHRR and HRC. Please discuss the mechanism.

The mechanism for the MP+PS system, for which the highest HRR and HRC reduction was achieved, was discussed in section “2.3.4. The mechanism of flame retardancy of developed FR system for Michael addition polymer”.

  1. “dehydration of phosphoric acid 400 were identified at ~1550 cm-1 and 1120 cm-1, respectively”. Please provide the references to support it.

We agree with the Reviewer's recommendation. As suggested, the references were added to the materials section.

  1. In addition to the Pyrolysis Combustion Flow Calorimeter, the LOI and UL-94 vertical burning tests are more essential to assess the fire retardant properties.

We fully agree with the Reviewer's comment and regret the lack of indicated analyses. This time, the problem is not equipment availability, but the material for deeper analysis. The raw materials for preparing the new polymer are quite expensive and were made as part of preliminary studies to apply for funding. For this reason, the authors wanted to study various properties to confirm the assumptions and determine the directions of further work. Currently, we do not have an adequate number of samples of the required dimensions to conduct further flammability tests. Thanks to the obtained funding, both studies, as well as tests using a cone calorimeter, will certainly be carried out according to schedule in order to study the mechanisms and select the best solutions.

Reviewer 2 Report

Comments and Suggestions for Authors
  1. The order of table captions is inconsistent—the first table is labeled as Table 3.
  2. The authors discussed the difference in particle sizes between SH and PS, despite undergoing the same grinding procedure. Given the broad particle size distribution, the reviewer is curious about the reproducibility of the data. For instance, do SH samples from three separate grinding batches show consistent particle size distributions?
  3. Does the larger particle size of SH influence the porosity of the resulting samples compared to those made with PS?
  4. The reviewer believes that the bio-sourced components do not significantly affect the material’s hydrophilicity, as the differences in contact angle appear to fall within experimental error.
  5. The DSC curve of the “APP+PS+SH+M” sample shows a small hump around 130 °C. Is this an instrumental artifact or a minor crystallization peak?
  6. The reviewer suggests including chemical structures or schematic illustrations of the compounds used in the synthesis. This would greatly aid readers in understanding the process.

Author Response

The answer to Reviewer 2:

We would like to thank the Reviewer for all his valuable remark. We agree with the recommendations that the manuscript should be improved, and so effort has been made to correct the article according to the comments.

  1. The order of table captions is inconsistent—the first table is labeled as Table 3.

The authors wish to thank the Reviewer for drawing attention to the error that was made.

  1. The authors discussed the difference in particle sizes between SH and PS, despite undergoing the same grinding procedure. Given the broad particle size distribution, the reviewer is curious about the reproducibility of the data. For instance, do SH samples from three separate grinding batches show consistent particle size distributions?

We thank the Reviewer for the remark. The grain size results can be considered representative in the authors' opinion. The same laboratory mill was used during grinding, the same method, and the same operator carried out the procedure. Then, all the batches after grinding were collected in one bag, because the amount of each plant component necessary to produce all the samples was relatively small. A representative batch was taken from each bag and subjected to grain distribution and microscopic analysis using SEM. The analysis was performed in three replicates, and the most representative result was presented in the article. The differences in the grain size distribution of these two raw materials are consistent with previous observations and regardless of the device used (doi:10.3390/polym11081234).

  1. Does the larger particle size of SH influence the porosity of the resulting samples compared to those made with PS?

Thank the Reviewer for valuable comment. The presented in the article porosity values represent their total volume, including the porosities in the interfacial region, but also contained in the filler it-self. Samples containing the flame retardant system with SH revealed the highest porosity. This effect is related to both the bimodal distribution of SH larger particle sizes, which leads to in-creased viscosity of the composition and difficulties in its degassing during forming, as well as the significantly higher porosity inside the particles. Respective changes have been taken into consideration.

  1. The reviewer believes that the bio-sourced components do not significantly affect the material’s hydrophilicity, as the differences in contact angle appear to fall within experimental error.

We thank the Reviewer for the suggestion. The authors have corrected the indicated sentence.

  1. The DSC curve of the “APP+PS+SH+M” sample shows a small hump around 130 °C. Is this an instrumental artifact or a minor crystallization peak?

Thank you for this comment. On this basis, the measurement was repeated for certainty, and the effect was noted again but with lower intensity. Because of the composition's complex structure containing plant derivatives, low-temperature stable extractives may degrade or be subjected to thermal conversion. The change in the DSC curve, as Reviewers noted, is probably endothermic. Moreover, similar to Tg, it is also observed on the DMA curves; hovewer, the values are shifted towards higher temperatures. Considering the relatively low intensity of the second effect observed on the damping factor curves, one can assume that this is an effect related to the occurrence of relaxation phenomena or post-hardening and reconstruction of the material structure. The question is justified; however, at the current research stage, it would be necessary to perform studies using coupled spectroscopic and thermal methods to describe this phenomenon. Thanks to the reviewer's comment, this research will be undertaken later in the current project.

  1. The reviewer suggests including chemical structures or schematic illustrations of the compounds used in the synthesis. This would greatly aid readers in understanding the process.

We thank the Reviewer for pointing that out. However, the structure of the polymers are described in our previous paper, citated as no 14 in literature (https://doi.org/10.3390/polym14194068). To avoid duplicating information, we have highlighted where you can find this information.

Round 2

Reviewer 1 Report

Comments and Suggestions for Authors

The authors have provided the responses to my concerned issues. I suggest it to be accepted in the current form.